# Zinc-Induced Folding and Solution Structure of the Eponymous Novel Zinc Finger from the ZC4H2 Protein

**DOI:** 10.3390/biom15081091

**Published:** 2025-07-28

**Authors:** Rilee E. Harris, Antonio J. Rua, Andrei T. Alexandrescu

**Affiliations:** Department of Molecular and Cell Biology, University of Connecticut, Storrs, CT 06269, USA

**Keywords:** HCA127, KIAA1166, C4 zinc finger, WRWF, Miles-Carpenter syndrome, glutaredoxin knuckle, protein folding, circular dichroism, molten globule

## Abstract

The *ZC4H2* gene is the site of congenital mutations linked to neurodevelopmental and musculoskeletal pathologies collectively termed ZARD (ZC4H2-Associated Rare Disorders). ZC4H2 consists of a coiled coil and a single novel zinc finger with four cysteines and two histidines, from which the protein obtains its name. Alpha Fold 3 confidently predicts a structure for the zinc finger but also for similarly sized random sequences, providing equivocal information on its folding status. We show using synthetic peptide fragments that the zinc finger of ZC4H2 is genuine and folds upon binding a zinc ion with picomolar affinity. NMR pH titration of histidines and UV–Vis of a cobalt complex of the peptide indicate its four cysteines coordinate zinc, while two histidines do not participate in binding. The experimental NMR structure of the zinc finger has a novel structural motif similar to RANBP2 zinc fingers, in which two orthogonal hairpins each contribute two cysteines to coordinate zinc. Most of the nine ZARD mutations that occur in the ZC4H2 zinc finger are likely to perturb this structure. While the ZC4H2 zinc finger shares the folding motif and cysteine-ligand spacing of the RANBP2 family, it is missing key substrate-binding residues. Unlike the NZF branch of the RANBP2 family, the ZC4H2 zinc finger does not bind ubiquitin. Since the ZC4H2 zinc finger occurs in a single copy, it is also unlikely to bind DNA. Based on sequence homology to the VAB-23 protein, the ZC4H2 zinc finger may bind RNA of a currently undetermined sequence or have alternative functions.

## 1. Introduction

ZC4H2 is a protein encoded by a gene on the X-chromosome with largely unknown functions. Mutations in the *ZC4H2* gene, primarily expressed in the brain and central nervous system during embryonic development, are associated with central and peripheral nervous system neurodevelopmental disorders and musculoskeletal diseases collectively called ZARD [1,2,3]. The most common of these, Wieacker–Wolff syndrome (WRWF), is an X-linked recessive arthrogryposis multiplex congenita disorder, with manifestations including muscular atrophy, neurodevelopmental impairment, ocular defects, and respiratory problems [1,4,5,6,7].

Some 90 interaction partners are listed for ZC4H2 in the IntAct database [8], with most of these identified from co-immunoprecipitation experiments. ZC4H2 is thought to bind to proteins in the SMAD pathway, affecting bone morphogenic protein signaling [9], which, together with transforming growth factor-β, regulates insulin transcription in pancreatic β-cells. These pathways have recently been implicated in a WRWF patient with hyperinsulemic hypoglycemia [10]. Another potential interaction target of ZC4H2 is the transient receptor potential cation channel subfamily V member 4 (TRPV4), a calcium ion channel that regulates cellular osmotic pressure. ZC4H2 is thought to bind to the cytosolic N-terminus of TRPV4, leading to increased basal activity and Ca^2+^ responses, together with increased channel turnover at the plasma membrane [11]. Notably, symptoms of ZARD resemble those of TRPV4-pathy, such as arthrogryposis, distal muscle weakness, club foot, and camptodactyly [12]. Finally, ZC2H2 has an intriguing possible role in proteostasis, modulating the Sonic Hedgehog (Shh) signaling pathway that governs embryonic spinal cord patterning and dendrite formation through glioma-associated oncogene (Gli) transcription factors [13]. ZC4H2 regulates the ubiquitination of RING finger protein 220 (RNF220), an E3 ubiquitin ligase enzyme that, by attaching the small 76 a.a. ubiquitin protein, epigenetically controls Gli expression gradients and effectively serves as an Shh signaling enhancer [14,15,16]. Interactions between ZC4H2 and RNF220 have been implicated in the development of noradrenergic neurons [15]. ZC4H2 also interacts with the E3 ubiquitin ligase RLIM, for which it itself is a substrate [17]. In partnership with the E3 ubiquitin ligases RNF220 and RLIM, ZC4H2 is poised to control protein levels post-translationally in several pathways involved in neural development. The interactions with the E3 ubiquitin ligases could form the basis of ZC4H2 dysfunction, in as much as mutations in RNF220 and RLIM lead to similar neurological pathology phenotypes as ZARD mutations [14,15]. To date there is no structural information on any protein interactions involving ZC4H2, or even what parts of the protein might participate in binding. Structural studies are therefore needed for a mechanistic understanding of ZC4H2 [17].

The 224 a.a. human ZC4H2 protein is predicted by the UniProt database (accession number Q9NQZ6) to consist of an α-helical coiled coil domain (residues 11–104) and a single putative novel ZnF domain (residues 189–206), from which the protein derives its name (Figure 1A). Immediately following the ZnF, is a putative nuclear localization signal (NLS) thought to run between residues 207 and 224 [18]. The *ZC4H2* gene was first identified in a human brain gene sequencing project, and initially named KIAA1166 [19]. Subsequently, the protein was found to be a human liver cancer antigen, named HCA127 [20]. The designation of ZC4H2 as a zinc finger protein is due to homology between HCA127 and an at-the-time novel sequence motif in the *C. elegans* epidermal morphogenesis regulator VAB-23 [21], described as a putative C4H2 ZnF. Fifteen years after the annotation of ZC4H2 [21], the ‘putative’ qualifier was dropped, but there is no published evidence that the sole globular domain in ZC4H2 or any of its homologs bind zinc or are functional.

Zinc fingers (ZnFs) are small 20–60 residue structural domains that fold upon binding one or more divalent zinc ions (Zn^2+^) [22,23,24,25,26,27]. About 3–5% of human proteins have ZnF domains, making this the most represented fold family in the human genome [23,28]. Because structures are largely stabilized by metal-binding, ZnFs are more structurally diverse than typical protein folds [27,29,30]. Over 50 ZnF subfamilies are known with cysteine (C) and histidine (H) Zn^2+^-ligand combinations such as CCHH, CCCH, CCHC, CCCC [26]. The CCHH DNA-binding family is the best characterized, occurrs in about half of all transcription factors [23,31]. While the CCHH ZnF family has a prototypical ββα fold (PDB: 1ZNF), other ZnF families have all-α (PDB: 1F81), all-β (PDB: 1K81), or folds lacking regular secondary structure altogether (PDB: 1PXE) [32]. The ZC4H2 domain does not have significant sequence homology to other sequence families; thus, the Prosite database (accession ID PDOC51896) classifies it as a distinct type of ZnF with no known structural representatives.

**Figure 1 biomolecules-15-01091-f001:**
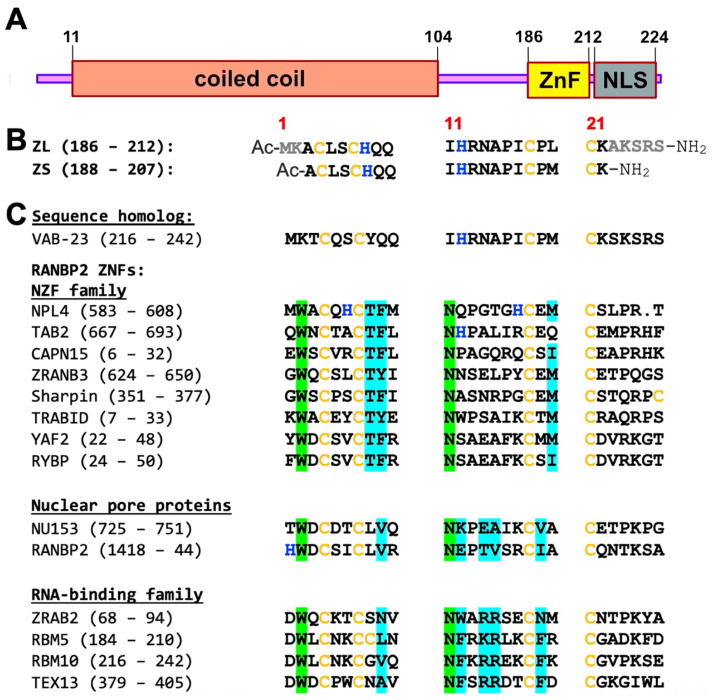
Domain organization and sequence of the ZnF from ZC4H2. (**A**) The human ZC4H2 protein consists of a coiled coil (orange) and a sole C-terminal ZnF domain (yellow) that is the subject of this work. A putative nuclear localization signal (NLS, gray) running from 207 to 224, overlaps with the C-terminus of the ZnF domain. (**B**) Sequence of the ZC4H2 ZnF. Initially we studied a short 20 aa fragment (188–207), ZS. Although ZS binds Zn^2+^, it gives NMR spectra similar to unfolded proteins. A longer 27 aa fragment (186–212), ZL, gives excellent NMR spectra and was used for structural studies. In this work, we use the 1–27 numbering scheme for the ZnF shown in red. (**C**) Sequence homologs, of which *C. elegans* VAB-23 is the closest. The family of RANBP2 ZnFs have a C4-ligand spacing and structure similar to ZC4H2-ZL. The RANBP2 NZF subfamily typically function as ubiquitin-binding modules in ubiquitination pathways [33,34]. NU153 and RANBP2 are nuclear pore proteins with protein-binding and possible DNA-binding functions [35,36]. A third RANBP2 subfamily has RNA-binding functions [37]. All sequences except VAB-23 are for human proteins and only sequences with a two-residue spacing between the first two cysteines are shown. Cysteines and histidines are colored yellow and blue, respectively. Residues highlighted in green are conserved in all RANBP2 homologs but not ZC4H2 or VAB-23. Residues highlighted in cyan are involved in ubiquitin-binding in the NZF subfamily [33,38] and nucleotide-binding in the RNA-binding subfamily [37].

NMR is ideally suited for determining the folding status and structure of ZnF domains. While it has been stated that cryoEM is the only structure determination technique worth investing in [39], typical ZnF domains at only 20–60 residues are too small for cryoEM. In multi-domain proteins, ZnFs are often separated from other domains by flexible linkers, so that their unconstrained rotational flexibility would lead to averaging of their electron densities into the noise. An exception is if the orientation of the ZnF domain were restricted by binding to other molecules, but it is unclear at this stage which, if any, part of ZC4H2 binds other proteins. We found it effective to use synthetic peptides to characterize ZnF domains, as this avoids the need for recombinant expression [40,41,42,43]. Here, we use synthetic peptide fragments to characterize the folding, metal binding, and structure of the novel ZnF domain from ZC4H2. We employ NMR and CD spectroscopy to show the ZC4H2 domain is a genuine ZnF that binds Zn^2+^ with high affinity. Because the ZnF has multiple Cys and His residues that are potential ligands for Zn^2+^, we use NMR pH titration of histidines and UV–Vis spectroscopy of a Co^2+^ complex to identify the four residues that chelate the metal. Having established the coordination site, we determine the NMR structure of the ZnF and compare it to functional families that share its metal-ligand spacing. Finally, we show evidence using two different-sized peptide fragments, that Zn^2+^ chelation precedes the formation of stable tertiary structure.

## 2. Materials and Methods

### 2.1. Samples

Fragments of human ZC4H2 (Figure 1B) were made by solid-phase peptide synthesis. The 27-residue peptide called ZL, corresponding to residues 186–212 was prepared to 97% HPLC purity by Biomatik (Kitchener, ON, Canada). A shorter 20-residue peptide ZS, comprising residues 188–207 was made to 90% HPLC purity by AAPPTec (Louisville, KY, USA). The differences in purity had no effect on folding or structure. Both peptides had acetylated N-termini and amidated C-termini to avoid the introduction of charges from free ends thus better mimicking the fragments in the context of the intact protein. Mass spectrometry of both peptides gave values within 2 Da of the theoretical masses of 2277 Da for ZS and 3066 Da for ZL. Peptide concentrations were determined using the BCA assay [44]. ZnCl_2_ (anhydrous, purity ≥ 98%), CoCl_2_·6H_2_O (purity ≥ 99%), and all other reagents were from Sigma (St. Louis, MO, USA). 

### 2.2. NMR Spectroscopy

To investigate zinc binding by NMR we looked at 1.6 mM samples of the ZS and ZL peptides at pH 6.0 and a temperature of 10 °C in the absence or presence of equimolar ZnCl_2_. All of these experiments were performed at 10 °C, as temperatures above 25 °C led to NMR line broadening and loss of amide proton signals. The samples had no other added buffers or salts. For NMR assignments and structure determination we used two samples of the longer ZL peptide in the presence of equimolar ZnCl_2_. A 1.6 mM sample of ZL-Zn^2+^ in 90% H_2_O/10% D_2_O at pH 6.0 was used to collect 2D TOCSY (70 ms mixing time), NOESY (150 ms mixing time) and ^1^H-^15^N HSQC (natural abundance) experiments in protonated solvent on a Bruker NEO 800 MHz spectrometer. A second 0.9 mM sample of ZL-Zn^2+^ in 99.8% D_2_O at pD* 5.9 was used to collect 2D TOCSY (70 ms mixing time), NOESY (50 ms mixing time) and multiplicity-edited ^1^H-^13^C HSQC (natural abundance) experiments in deuterated solvent on a Bruker NEO 600 MHz spectrometer. Both spectrometers were equipped with cryogenic probes. Internal DSS (2,2-dimethyl-2-silapentane-5-sulfonate) was used for ^1^H and ^13^C referencing, while an indirect Ξ (^15^N/^1^H) ratio of 0.101329118 was used for ^15^N referencing [45]. We obtained >80% sidechain assignments and >92% backbone assignments for ^1^H, ^13^Cα and ^15^N resonances, but the ^15^N assignments should be considered tentative particularly for weak resonances, due to the low sensitivity of the ^1^H-^15^N HSQC spectrum at natural isotope abundance.

Additional NMR experiments were performed to study the biophysical properties of ZL-Zn^2+^. A pH titration of 0.9 mM ZL-Zn^2+^ in D_2_O at a temperature of 10 °C was used to investigate if histidines were involved in Zn^2+^-binding. The pH data were fit to a modified Henderson-Hasselbach equation to obtain histidine p*K*_a_ values [46]. DOSY spectroscopy [47] to verify the monomeric state of ZL-Zn^2+^ was performed on a 0.3 mM sample in D_2_O, at pD* 6.9 and a temperature of 25 °C. A temperature titration to assess the thermal stability of ZL-Zn^2+^ was performed on a 0.3 mM sample in D_2_O, at a pD* of 6.9. Data up to 50 °C were collected on a 600 MHz Varian Inova spectrometer but data at higher temperatures were collected on a 500 MHz Bruker Avance instrument, since our Varian probe did not allow work above 50 °C. NMR data were processed with iNMR (http://www.inmr.net/) and analyses were performed with the program CcpNmr Analysis 2.5.2 [48].

### 2.3. CD Spectroscopy

CD experiments were performed on an Applied Photophysics Chirascan V100 Spectrometer (Surrey, UK) using a 1 mm cuvette path-length, a 1 nm bandwidth, a 1 nm scan step size, and 5 s/point data averaging. ZS and ZL sample concentrations were 40 and 80 µM, respectively. The samples were in 5 mM HEPES buffer (pH 6.8 to 6.9), containing 0.2 mM of the reducing agent TCEP (tris (2-carboxyethyl)phosphine) to prevent cysteine disulfide formation. To measure metal binding affinity, ZnCl_2_ titrations at a temperature of 22 °C were performed in the presence of 10 mM competitive chelator EGTA [42,49]. The *K*_d_ was calculated from the EGTA competition experiments as previously described [42].

### 2.4. UV–Visible Spectroscopy of Co^2+^ Complexes

Spectra as a function of CoCl_2_ were recorded from 200 to 800 nm on an Ultrospec 8000 double-beam spectrophotometer (Thermo Fisher, Waltham, MA, USA) using 500 µL samples in 750 µL cuvettes with a 1 cm pathlength at a temperature of 25 °C. The ZS and ZL sample concentrations were 390 and 200 µM, respectively. Peptides were buffered at pH 7.0 in 10 mM TRIS, containing 0.5 mM TCEP. Samples were blanked against cuvettes containing buffer alone.

### 2.5. Alpha Fold 3 Simulations of Random Sequences

Five random sequences each, for protein lengths between 20 and 150 in increments of 10 residues, were generated with the random sequence server of EXPASY (https://web.expasy.org/randseq/ accessed on 22 May 2025). The sequences were submitted to the Alpha Fold 3 (AF3) server (https://alphafoldserver.com/ accessed on 22 May 2025) to generate structure predictions [50]. For random sequences matching the cysteine and histidine composition of the ZnF from ZC4H2, we generated 25 random sequences with the sequence length set to 27 residues, and specified cysteine and histidine compositions of 14.8% and 7.4%, respectively. The latter sequences were submitted to AF3 for prediction with inclusion of a single Zn^2+^ ion. All of the random sequences used for this work are available from the corresponding author upon request.

### 2.6. NMR Structure Calculations

Dihedral angle restraints were calculated from assigned chemical shifts using the program TALOS-N v 4.21 [51]. Distance restraints from NOESY spectra were set to three upper ranges of 3.0, 4.0 and 5.0 Å, respectively, based on peak intensities. The lower bound was uniformly set to 1.8 Å. Pseudo-atom upper bound corrections of 1.0, 2.0, 2.4 and 1.5 Å were included for protons with ambiguous methylene, aromatic ring, prochiral methyl, and methyl group NMR signals, respectively [52]. Five hydrogen bond restraints (four β-hairpin and one α-helix) were included based on the secondary structure consensus, and proximity of hydrogen bond donors and acceptor atoms in initial NMR structures excluding hydrogen bonds. The Zn^2+^ atom and ligands were restrained using distance bounds of 2.33–2.37 Å for Zn^2+^-Sγ and = 3.25–3.51 Å for Zn^2+^-Cβ [41] for each of the four cysteines. An additional six restraints of 3.02–4.52 Å were included between each pair of cysteine Sγ atoms to enforce the tetrahedral geometry of the binding site [53]. The PdbStat v 5.10 program was used to remove redundant and structurally non-informative restraints [54].

NMR structure calculations were initially performed on the NMRbox platform [55] with the program X-PLOR NIH v. 3.8, using the protein-4.0 parameter set [56]. Structures were calculated using distance geometry, followed by simulated annealing refinement using the X-PLOR script prot_sa_refine_nogyr.inp from the NESG (https://nesgwiki.chem.buffalo.edu accessed on 20 May 2025). Once we obtained restraint sets without violations, we further water-refined the structures using the program ARIA 2.3.2 [57] on NMRbox. A set of 200 structures in explicit H_2_O solvent were calculated with the default iteration protocols of the ARIA program, from which the 20 structures with the lowest energies and no violations were chosen for deposition and analyses (Appendix A).

## 3. Results

### 3.1. ZC4H2 Has a Genuine ZNF with Domain Boundaries Larger than Those Specified by UniProt

The UniProt database predicts the ZC4H2 protein comprises a coiled coil, followed by a single C-terminal ZnF domain with a unique C-X2-C-H-X3-H-X5-C-X2-C sequence pattern (Figure 1A,B). Since no members of the ZC4H2 homology family have yet been shown to be genuine ZnFs, we first wanted to see if the domain from the human protein folded in the presence of Zn^2+^. We initially studied a synthetic peptide fragment extended by one amino acid on either side of the 189–206 domain boundary specified by the UniProt database (accession code Q9NQZ6). We call this short 20 a.a. segment (188–207) ZS (Figure 1B). Although there are differences between the 1D NMR spectra of the peptide with and without Zn^2+^ (Figure 2A, ZS), these are small, and the amide proton NMR signals largely fall in the random coil region of the spectrum. The 2D TOCSY spectrum of ZS similarly has random coil-like chemical shifts, as well as NMR line broadening characteristic of dynamics on the µs-ms timescale (Figure 2B, ZS-Zn^2+^). Despite the poor NMR spectra, the ZS fragment undergoes changes in CD spectrum characteristic of a folding transition with increasing Zn^2+^ concentration (Figure 3A). Using a previously described assay measuring CD signal changes upon Zn^2+^ addition in the presence of the competitive chelator EGTA [43,49], we calculate the ZS fragment binds Zn^2+^ with a *K*_d_ of 18 pM (Figure 3C).

We suspected that the ZS fragment gave poor NMR spectra because its domain boundaries were too short, disturbing the structure of the ZnF (Figure 1B). We consequently looked at a longer 27 a.a. fragment encompassing residues 186–212 of ZC4H2, called ZL. Compared to ZS, the ZL fragment extends the sequence by two residues at the N-terminus (Figure 1B) that are involved in the β-hairpin structure of ZnF homologs (Figure 1C). AF3 predictions also suggest that the ZS fragment puts the N-terminus in the middle of the first β-strand while the longer ZL fragment has the first β-strand intact. We also extended the C-terminus with the polar residues AKSRS (Figure 1B), expecting this would improve solubility. The ZL fragment shows several backbone and side chain amide protons outside the random coil region of the spectrum (labeled in Figure 2A,B), and while the fragment lacks any aromatic residues other than two histidines, it gives excellent NMR chemical shift dispersion. DOSY spectra [47] show the Zn^2+^-bound ZL fragment is a monomer (Appendix A).

Although line broadening affects NMR signals from ZL at temperatures above ~25 °C, NMR spectra improve at a temperature of 10 °C. The good chemical shift dispersion of the ZL fragment (Figure 2A, ZL-Zn^2+^) allowed us to obtain NMR assignments for the entire sequence (Appendix A). The differences between the ZL and ZS NMR spectra are not due to amide proton hydrogen exchange. Non-exchangeable carbon-bound hydrogens in the aliphatic region of the NMR spectra also give increased line broadening and lowered chemical shift dispersion for ZS compared to ZL (Appendix A) suggesting the shorter fragment is subject to dynamic heterogeneity due to its inability to form a stable structure. Despite the superior NMR spectra for ZL, the changes in the CD spectrum of ZS with Zn^2+^ are similar to those for the shorter ZS fragment (Figure 3A,B). Moreover, the *K*_d_ of 18 pM for Zn^2+^ for the ZL fragment is the same as that for the ZS fragment (Figure 3C,D). This suggests the Zn^2+^ binding site as detected by CD is similar in the ZS and ZL fragments, but only the longer ZL fragment achieves a stable tertiary structure as monitored by NMR.

Compared to other ZnFs we recently studied that have thermal melting points > 80 °C [40,42,43], ZC2H4-ZL undergoes a thermal unfolding transition by NMR at a much lower temperature of 32 ± 6 °C (Appendix A). Broadening of NMR resonances and loss of amide peaks at high temperatures is why we did all structural studies with ZC4H2-ZL at a temperature of 10 °C. Although the changes in the NMR spectrum with temperature are completely reversible, the NMR signals from the thermally unfolded state are broad (Appendix A, Uε1 and Uδ2), suggesting NMR relaxation contributions from motion on a µs-ms timescale typical of molten globules [58,59], or indeed the ZS fragment (Figure 2A). This is not the case with the protein unfolded by acidic pH, where the NMR signals from the unfolded state are sharper than those from the folded state (Figure 4B, Uε1 and Uδ2). These observations raise the possibility that the thermally unfolded state may still have zinc bound. We note the thermal unfolding transition observed by NMR occurs near physiological temperatures, but do not know if this has functional significance.

### 3.2. The ZnF of ZC4H2 Has a CCCC Metal Coordination Sphere

NMR data for the ZL fragment appeared amenable to structure determination. However, ZnFs such as ZC4H2 with multiple potential metal ligands can confound the identification of metal-coordinating residues [42,43]. The four cystines and two histidines in ZL can be arranged in 11 different coordination combinations to bind a single zinc atom (1 CCCC, 4 CCCH, and 6 CCHH). Considering a four-cysteine coordination set, the C-X2-C-X10-C-X2-C sequence spacing in ZC4H2 is similar to the RANBP2 family [34,37] with a C-X(2-4)-C-X10-C-X2-C sequence pattern (Figure 1C) but also to the N-terminal part of the C6 ZnF from lysine-specific demethylase hairless (UniProt O43593, a.a. 600–625) which has a C-X2-C-X10-C-X2-C-X4-C-X2-C sequence pattern. The underlined portion of the sequence is the part similar to ZC4H2-ZnF. Other similar cysteine spacings are found in parts of ZnFs that bind two zinc ions, such as the first half of the MYND domain: C-X2-C-X(7-11)-C-X2-C-X5-C-X3-C-(X7,8)-H-(X3)-C (Uniprot Q8IYR2, a.a. 296–335) and the second half of ring fingers such as MGRN1: C-X2-C-X(9-39)-C-X(1-3)-H-X(2-3)-C-X2-C-x10-C-X2-C (Uniprot O60291, a.a. 278–317). An important prerequisite for structure determination is to first accurately identify the residues that participate in zinc coordination.

To this end we obtained NMR pH titration data for the ZL fragment. The rationale is that histidines bound to zinc should be invariant to changes in pH when they are bound to zinc [42,61]. By contrast, histidines not involved in metal coordination should titrate freely with pH as they undergo their protonation-deprotonation equilibria. Starting from our complete backbone NMR assignments, we first assigned the aromatic ring protons of the two histidines through weak four-bond ^4^J_Hδ2-Hβ#_ couplings observed in the 70 ms TOCSY spectrum for the sample in D_2_O (Figure 4A). The data for the aromatic region of the ZL NMR spectrum (Figure 4B) indicates both histidines titrate freely between pH 4.0 and 9.5, suggesting they are not involved in zinc coordination. Rather, the four cystines in ZC4H2 must form the zinc coordination site. H8 has a p*K*_a_ of 6.9 and H12 has a p*K*_a_ of 5.5 in the folded zinc-bound state (Figure 4C). For comparison, both histidines give p*K*_a_ values near 6.0 in the unfolded state without zinc (Appendix A). The shift in p*K*_a_ values between the zinc-free and zinc-bound ZL is due to the change in environment of the histidines accompanying metal-induced folding of the peptide [46,62,63,64].

Metal coordination was further studied using a CoCl_2_ titration of the ZL fragment monitored by UV–Vis spectrophotometry (Figure 4D). Cobalt is a structural analog of Zn^2+^ that is colorimetrically active and provides valuable information on the metal coordination site [24,60,65]. The ligand-metal-charge-transfer (LMCT) band at 315–340 nm is sensitive to the number of thiolate groups bound to cobalt, with each cysteine contributing about ~1000 M^−1^ cm^−1^ to the extinction coefficient [60]. The Co^2+^ titration data for ZL, with an LMCT ε_315_ of about 4000 M^−1^ cm^−1^ supports the coordination of a single cobalt ion by four cysteines (Figure 4D). The d-d transition bands between 500 and 800 nm are sensitive to coordination geometry, with octahedral, pentacoordinate, and tetrahedral biding sites giving extinction coefficients below 30 M^−1^ cm^−1^, between 50 and 250 M^−1^ cm^−1^, and above 300 M^−1^ cm^−1^, respectively [24,60]. The d-d transition bands for both the ZS and ZL fragments with ε_700_ > 400 M^−1^ cm^−1^ are consistent with a tetrahedral coordination sphere for Co^2+^ (Figure 4D, inset). Moreover, the d-d transition bands are red-shifted, supporting a metal coordination site consisting exclusively of thiol groups [66,67]. The inset compares the d-d band spectrum for the fragments (Figure 4D). Despite differences in the quality of NMR spectra for the ZL and ZS fragments (Figure 2), the similarity of their UV–Vis spectra suggest they both bind Co^2+^ using similar tetrahedral CCCC-coordination modes.

### 3.3. Alpha Fold 3 Predicts the ZnF Structure with Confidence but Also Similarly Sized Random Sequences

The AF3 program [50] predicts a structure for the ZL fragment consisting of two orthogonal hairpins, with each of the hairpins contributing two cysteines to the zinc binding site (Figure 5A). The structure is typical of the RANBP2 family of ZnFs [34,37], a metal-binding folding motif also found in iron-binding proteins such as rubredoxin and hence referred to as the ‘rubredoxin knuckle’ [34]. In the AF3 model, the four cysteines in the sequence coordinate Zn and the histidines do not participate, consistent with our results from pH titrations and UV–Vis spectroscopy of cobalt complexes (Figure 4). Given that AF3 confidently predicts a structural model for the ZnF of ZC4H2, a reasonable question is whether it is worthwhile to calculate an experimental NMR structure, a process that is considerably more time-consuming and laborious than an AI-generated structure prediction. As outlined below we feel that even if the AF3 prediction is trustable, it is important to verify this experimentally for targets such as the ZnF of ZC4H2 that have no representative structures.

While the two-hairpin motif is predicted for the ZL fragment (Figure 5A), for the shorter ZS fragment AF3 predicts a loop structure in which both the individual hairpins are lost (Figure 5B). Note that there is also a switch between the positions of the two zinc ligands C189 and C192 (relative to C203 and C206) in the two models (Figure 5A,B). AF3 calculates a model similar to that in Figure 5A when the ZL prediction is performed without zinc, but with the shorter ZS fragment a loop linked by two disulfide bonds is predicted in the absence of zinc (Figure 5C). All models have light or dark blue colors corresponding to confident or very confident predictions (pLDDT scores better than 70). Yet there are differences in the predictions depending on the lengths of protein fragments.

To gain further insight into the confidence levels of AF3 predictions for domains similarly sized to the ZnF of ZC4H2, we tested how prediction confidence varied with increasing sequence length. We used the RandSeq server of ExPASy [68] to generate five random sequences for each polypeptide length between 20 and 150 residues, in increments of 10 amino acids (Figure 5D). The majority of the AF3 predictions had substantial and realistic-looking secondary structures. For sequences longer than 100 residues, the pLDDT scores were predominantly below 50, corresponding to low-confidence predictions as expected for random amino acid sequences. For sequences shorter than 100 amino acid, however, the predictions increased in confidence with decreasing sequence length.

The data in Figure 5D consider the overall pLDDT score for the prediction. In nearly all groups of five predictions shorter than 100 residues, at least one of the five predictions had a segment with a pLDDT score better than 70, even when the protein had an overall lower score. These confident predictions could correspond to structured domains or subdomains in proteins that are otherwise unfolded, and this is not accounted for in the statistics of Figure 5D. We are not sure why AF3 confidently predicts apparent false positive structures for proteins shorter than ~50 amino acids with random sequences. This could be due to the spurious ‘hallucination’ of structures for disordered regions that was noted as a limitation for the diffusion-based AI algorithm for AF3 [50]. Alternatively, the occurrence of high-confidence predictions for short random sequences could be a type of “Russian doll effect” [69], where shorter sequences are statistically more likely to match a sequence in the X-ray structure database used for AI training. Since AF3 cannot distinguish structured from disordered polypeptides, the occurrence of a reasonable match in the X-ray database raises the confidence of the prediction, even when the matching short segment would likely be unfolded outside the context of the larger protein.

We saw similar behavior with a set of 25 random sequences selected to match the 27-residue length and 4 cysteine plus 2 histidine content of the ZnF in ZC4H2. Five representatives of these random sequences are given in Figure 5E and their corresponding AF3 predictions in Figure 5F,G. Statistics pertaining to pLDDT confidence levels for the predictions, and to the number of Cys/His coordinating the zinc ion are summarized in Figure 5H and Figure 5I, respectively. Most of the 25 random sequences in the trial dataset are confidently predicted (88% pLDDT > 70) with four giving very confident predictions (16%, pLDDT > 90). These observations are consistent with the behavior of 30-residue polypeptides in the larger dataset of random sequences (Figure 5D). Note that the observed pLDDT confidence scores for the random sequences are of the same order as those for the prediction of the actual sequence for the ZnF in ZC4H2 (Figure 5A–C). Remarkably, the majority of ZC4H2-like random sequences (52%) have a combination of four Cys/His residues (green in Figure 5G) within bonding distance of the zinc ion, with a further 40% having more than four Cys/His near the metal, and 8% having only three Cys/His in the coordination site. While rare, folded ZnFs with only three metal ligands coordinating zinc occur in nature [42]. Thus, the random ZnF-like sequences have a high chance to give realistic-looking zinc coordination sites in AF3 predictions (Figure 5I).

Small proteins, including ZnF domains, are often viewed as trivial “low-hanging fruit” in current structural biology. However, it is becoming increasingly appreciated that there is a vast repertoire of thousands of biologically important mini and microproteins shorter than 50–100 amino acids [70,71] who’s structural properties are largely unexplored. Given the ambiguities of AF3 predictions for such small proteins, these mini and microproteins could represent a future frontier for experimental structural biology. As funding priorities increasingly shift from curiosity-driven science to AI, technology, and for-profit research, these structures may unfortunately remain uncharted at least in the short term.

### 3.4. The NMR Structure of ZC4H2 Is a Variant of the RANBP2 ZnF Fold

The experimental NMR structure of the ZL fragment corresponding to the ZF in ZC4H2 is shown in Figure 6A,B. The NMR structures are shown on a color ramp from the N- (blue) to the C-terminus (red). The folding motif has similarity to the RANBP2 ZnF fold, where two orthogonal glutaredoxin knuckle β-hairpins each provide two cysteines to form a coordination cage for zinc [34,37]. Conceptually, this fold replaces the α-helix in the classical ββα motif of CCH (H/C)-type ZnFs [23,43] with a second β-hairpin to give a ββ-ββ structure. In the ZnF of ZC4H2 the second hairpin lacks canonical β-sheet secondary structure since there are two prolines that preclude β-sheet hydrogen bonding. Other ZnFs with glutaredoxin knuckles such as TAB2-NZF [72] and the RANBP2 ZnF from FUS [73] also do not have β-sheet structure for the second hairpin. A unique feature of the ZnF in ZC4H2, is the presence of a short one-turn α-helix at the end of the second hairpin (Figure 6A). This proved difficult to identify as it is not predicted by AF3 (Figure 5A) but is supported by α-helical chemical shift indices as well as short range NOEs characteristic of α-helical structure for residues C21-S25 at the C-terminus of ZL (Appendix A). Consequently, the RMSD between the NMR structure and the AF3 model is rather large at about 4 Å. Predictions are typically considered accurate when the RMSD is below a threshold of about 2.5 Å [74].

Although the short α-helix is present in most NMR conformers for the ZL fragment, its relative position is less well defined than the remainder of the structure (Figure 5B). The chemical shift indices predict a gradient of increasing disorder for the last four residues at the C-terminus, with the last two residues giving chemical shift-predicted S^2^ order parameters below 0.5. We included the last four residues 24-KSRS-27 to improve the solubility of the ZL fragment. The highly polar character of these amino acids probably accounts for the disorder observed at the C-terminus of the NMR structure.

## 4. Discussion

### 4.1. Structural Mapping of the Eight ZARD Mutations Within the ZnF

Our work establishes the presence of a genuine folded ZnF in ZC4H2. We can use our NMR structure (Figure 6C) to analyze the locations of nine ZARD mutations distributed amongst seven sites in this ZnF domain: M186R, R198Q, A200T, A200V, P201H, P201S, C203S, C206F and R211W [2,3,10]. The M186R mutation is in the first hairpin and may disturb its β-sheet secondary structure. C203S and C206F substitute non-chelating residues for the third and fourth zinc ligands. The A200T and A200V mutations introduce bulkier sidechains that could lead to steric clashes between the two hairpins in the structure, particularly between the A200 substitutions and I196 from strand β1. The P201H and P201S mutations may interfere with the ability of P201 to form a turn that orients the second hairpin at a right angle to the first. Two mutations R198Q and R211W have no obvious structural roles, and as such probably replace residues important for molecular recognition functions in ZC4H2. Of these two sites, R211 is at the junction between the ZnF and the subsequent NLS used to direct the protein to the cell nucleus [18,21]. A tenth mutation R213W, also in this region interferes with nuclear transport [4,18].

### 4.2. Hierarchical Folding of the ZnF Domain in ZC4H2

Our results show that a 20-residue segment, ZS, is sufficient to form a high-affinity zinc-binding site (Figure 3), but a longer 27-residue segment, ZL, is needed to obtain specific tertiary sidechain packing interactions (Figure 2), suggesting the motif folds hierarchically. When the ZL fragment is thermally unfolded near physiological temperature (Appendix A), the resulting molten globule may still have zinc bound. There is precedent in the literature for sequential non-two-state folding of ZnFs. Chelation of zinc by CCHH-type ZnFs is thought to occur in a stepwise manner with the cysteines that form the strongest interactions with the metal binding first, followed by the N-terminal His and finally the C-terminal His. An initial hydrophobic collapse is followed by the formation of the cysteine containing β-hairpin component of the ββα fold, with final stabilization by the α-helix that contains the histidines [75,76]. Similarly, there is evidence for biding of zinc by a CCCC tetradentate coordination site in the mitochondrial protein TIM10 that first involves the formation of an unstructured encounter complex as monitored by fluorimetry, followed by the development of folded structure at higher zinc concentrations in the final complex as monitored by CD [49]. Moreover, several zinc fingers show increased protein dynamics in mutant forms giving NMR spectra that approach those of unfolded proteins but retain the ability to bind zinc and adopt an overall folded structure [42,77,78].

To examine why the ZS fragment gives NMR spectra typical of unfolded proteins but retains CD spectra consistent with folded secondary structure (Figure 3A), we did a simulation in which we calculated NMR structures keeping only dihedral restraints and restraints to zinc. There is evidence that backbone dihedral angles could be dictated by short-range interactions between residues separated by only a few positions in the polypeptide sequence [79]. When all NOE distance restraints were removed, to simulate a molten globule conformation [80], in which secondary structure is retained but sidechain interactions in the non-polar core of the protein become disordered, the resulting NMR ensemble shows a considerable spread of conformations giving a backbone of RMSD of 3.9 Å for the bundle. Nevertheless, the overall N- to C-term topology of the chain is retained (Figure 6D) albeit with the loss of the N-terminal β-hairpin structure. The result could explain why the ZL and ZS fragments share similar far-UV CD spectra characteristic of polypeptides with folded secondary structure when zinc is bound, even though the NMR spectra suggest only ZL has specific tertiary structure. In a further simulation we removed all dihedral and distance restraints and kept only restraints between the four ligating cysteines and zinc. Even in this case, the N- to C-terminal topology of the chain is retained, although the spread in conformations is larger, giving an RMSD of 4.7 Å (Figure 6E). The simulation in Figure 6E is probably unrealistic since a substantial number of residues move into disallowed regions of Ramachandran space. Taken together, these observations highlight that zinc fingers can fold sequentially, with zinc coordination forming a nidus that provides a platform for more extensive structure development.

That ZnFs fold in a hierarchical manner has implications for assessing the functions of these modules. Our approach for assessing the viability of ZnFs is to make synthetic peptide fragments corresponding to the domains, to see if these give NMR spectra typical of folded structures in the presence of zinc. Some novel ZnFs were characterized in this way, and we were able to determine that several ZnFs annotated as degenerate by UniProt were in fact functional [40,41,42,43]. The results with the ZS fragment show that in some cases folding upon zinc binding could be missed by NMR which is more sensitive to fixed tertiary structure, compared to far-UV CD which monitors secondary structure. The Protein Structure Initiative used up most resources in structural biology during the period 2000–2015 with the stated goal of cataloging every type of protein structure [81]. Part of this effort involved the development of automation to identify the most suitable candidates for structural studies. It is worth considering how many interesting ZnFs and other proteins could have been missed by this approach because protein dynamics or other factors conspired to thwart the ^1^H-^15^N HSQC or crystal quality standards of that time.

### 4.3. The ZnF of ZC4H2 Represents a Unique Sequence Family Amongst Structural Homologs

The C4 coordination site sequence pattern we established for ZC4H2 is also found in the RANBP2 family of ZnFs (Figure 1C). Additionally, RANBP2 ZnFs have a related structure consisting of two orthogonal β-hairpins [33,34,37]. The prototypical member of this ZnF family, RANBP2, is a large nuclear pore protein with multiple zinc fingers that bind to the small GTPase protein Ran, which regulates the protein’s nuclear import functions [36,82]. There are hundreds of members of the RANBP2 ZnF family [83], but their functions fall into three major categories: the ubiquitin-binding NZF subfamily [33], nuclear pore proteins [36,84], and the RNA-binding RANBP2 family [37].

The RANBP2-like family has the sequence signature W-X-C-X(2,4)-C-X3-N-X6-C-X2-C [83]. The ZnF in ZC4H2 shares this cysteine spacing but lacks the conserved tryptophan and asparagine at positions 2 and 11 of this pattern (Figure 1C). We used the MOTIF2 server (https://www.genome.jp/tools/motif/MOTIF2.html accessed on 27 April 2025) to search for alternative homology relationships for the ZnF of ZC4H2. If we constrained the search to include the two histidines in addition to the four cysteines with their appropriate spacing: C-X2-C-H-X3-H-X5-C-X2-C, the only hit is the ZC4H2 family. If we included the cysteines and the two unusual prolines, C-x(2)-C-x(8)-P-x-C-P-x-C, in addition to ZC4H2 we obtained hits to three RING finger families (LON peptidase 1, LON peptidase 2, and deltex E3 ubiquitin ligase 3L). While it is possible that the ZC4H2-ZnF might be evolutionarily derived from a RING finger, the homology is unlikely to be functionally relevant. This is because the RING finger motif has 8 coordinating residues that bind to two zinc ions [85]. In all three cases, the hits were to the second half of the RING finger sequence, where the first two cysteines bind to one zinc ion and the last two to the other [85]. Motif searches incorporating the conserved cysteines and residue types other than the prolines failed to find ZnF families other than ZC4H2 and the closely related homolog VAB-23. Thus, the ZnF domains from ZC4H2 and VAB-23 appear to have unique sequence properties amongst ZnFs, with the RANBP2 family being the closest matches in terms of the structure. In addition to a distinct sequence, the structure of ZC4H2-ZnF differs from RANPB2 ZnFs by ~4 Å RMSD and has a non-conserved C-terminal α-helix.

The best characterized ZnFs are CCHH-types that occur as DNA-binding modules in transcription factors. Yet on their own, these modules lack the ability to recognize specific dsDNA sequences, which typically requires at least two to three ZnF modules [86]. While C4 ZNFs are known to bind DNA, examples such as nuclear receptors involve a complicated sequence pattern that binds two zinc atoms to form an α-helical structure [87]. It has been reported that RANBP2 ZnFs in nuclear pore proteins such as RANBP2 and NU153 bind DNA, but the fragments studied involved eight and four ZnF modules, respectively. No information was reported on DNA sequence specificity, or binding affinities for the nuclear pore proteins [36,84]. Up until recently there were no known cases of a single C4 ZnF that binds DNA, but two examples were found in *Myobacterium smegmatis* [88]. These were 179 and 216 residue proteins, however, not the 27-residue domain found in ZC4H2. Unless it represents an unprecedented prototype, we feel it is unlikely the ZnF in ZC4H2 has a DNA-binding function since it occurs in a single copy.

A second branch of the RANBP2 family termed NZF, functions to bind ubiquitin in various post-translational modification pathways [33]. The possibility that ZC4H2-ZnF could bind ubiquitin like its NZF structural homologs was intriguing in light of the literature showing ZC4H2 interacts with the E3 ubiquitin ligases RNF220 [14,15,16] and RLIM [17] to control several pathways involved in neural development [14]. Moreover ubiquitin-binding NZF domains usually occur at the N- and C-termini of proteins, like the ZnF in ZC4H2 [33]. We tested if the ZL fragment interacts with ^15^N-labeled human ubiquitin but detected no binding (Appendix A). In retrospect, this is not surprising, since compared to the NZF sequence family, the ZnF of ZC4H2 lacks the key residues T8 and F9 (Figure 1C) that are critical for ubiquitin binding [33,34].

A final class of RANBP2-type ZnFs have RNA-binding functions. Using a SELEX approach it was shown that several RANBP2-type ZnFs including those from the proteins ZRANB2, EWS, RBP56, RBM5, and TEX13A, form a 1:1 complex with 6–17 nucleotide fragments of ssRNA containing the consensus sequence AGGUAA [37,89]. A RANBP2-type ZnF in the protein FUS, recognizes the related consensus sequence NGGU [73]. Of these proteins, the testis-expressed protein TEX13A is particularly intriguing since it has the same domain organization as ZC4H2, consisting of a long coiled coil followed by a ZnF. However, an examination of the residues in the RANBP2 ZnFs that form key structural interactions with RNA bases [37] shows these are not conserved in ZC4H2 (cyan in Figure 1C). Thus, while it is possible that ZC4H2 has an RNA-binding function, the nucleotide sequence of the putative RNA must be different than those bound by RANBP2 ZnFs.

Perhaps the strongest clues about the function of the ZnF in ZC4H2 come from its closest homolog, the *C. elegans* protein Vab-23. Homology for the two proteins is particularly high for the ZnF and subsequent NLS regions (Figure 1C), which were originally used to infer the presence of a ZnF motif in ZC4H2 [21]. Vab-23 is a regulator of epidermal morphogenesis in *C. elegans* embryos and its ZnF domain is required for this function. The protein is localized to nuclear speckles in a manner consistent with pre-mRNA processing or mRNA regulation [21] suggesting it binds RNA. Thus, an RNA-binding role for the ZnF of ZC4H2 is plausible, albeit with RNA sequence determinants different from those of the RANBP2 domains. Alternatively, since ZC4H2 has both unique sequence and structural properties it may have an alternative function.

## 5. Conclusions

In this work we showed that ZC4H2 contains a genuine ZnF which has implications for the structure and function of the protein, as well as its role in disease. Establishing the presence of a genuine ZnF in the ZC4H2 protein was not trivial, since the AI-based AF3 program predicts folded structures for peptides of similar lengths with random sequences that are unlikely to be folded. Even using experimental methods, establishing the presence of a functional ZnF proved difficult since the UniProt-specified domain boundaries gave a peptide that binds zinc but forms a molten globule, lacking specific sidechain packing until the fragment is appended at its N- and C-terminal ends. Thus, testing the viability of a ZnF can require multiple experimental techniques as well as optimization of domain boundaries to achieve proper folding. The presence of a ZnF in ZC4H2 raises the possibility that this domain could participate in its protein–protein interactions, including with SMAD proteins [9], the TRPV4 ion channel [11], and the RNF220 and RLIM E3 ubiquitin ligases [17], amongst others.

Nine ZARD mutations occur in the ZnF and while most of these are likely to perturb the structure, at least one mutation, R198Q, occurs at a position that does not appear to serve a structural role. As such the mutation could disrupt molecular interactions involving the ZnF. Two other mutations R211W and R213W near the C-terminus also do not appear to have structural roles but overlap with a NLS sequence [18] and could affect binding interactions or nuclear import of ZC4H2.

The sequence and NMR structure of the ZC4H2 ZnF are similar to those from the RANBP2 family but have important differences in sequence and structure. Because the ZnF modules share similar ligand spacing we looked for functional commonalities. Unlike the NZF branch of the RANBP2 family the ZC4H2 ZnF does not bind ubiquitin. The ZC4H2 ZnF is unlikely to bind DNA since it occurs as a single copy and is probably too short to provide an epitope for specific DNA-binding [86,88]. It is possible that ZC4H2 ZnF binds RNA since it has a high proportion of positively charged residues and a fold used for this function in other ZnFs, but the RNA sequence-specificity must be different than that for the RNA-binding branch of the RANBP2 family since key binding residues are missing. The current structural work sets a foundation for determining the function of the ZnF in the ZC4H2 protein.

## Figures and Tables

**Figure 2 biomolecules-15-01091-f002:**
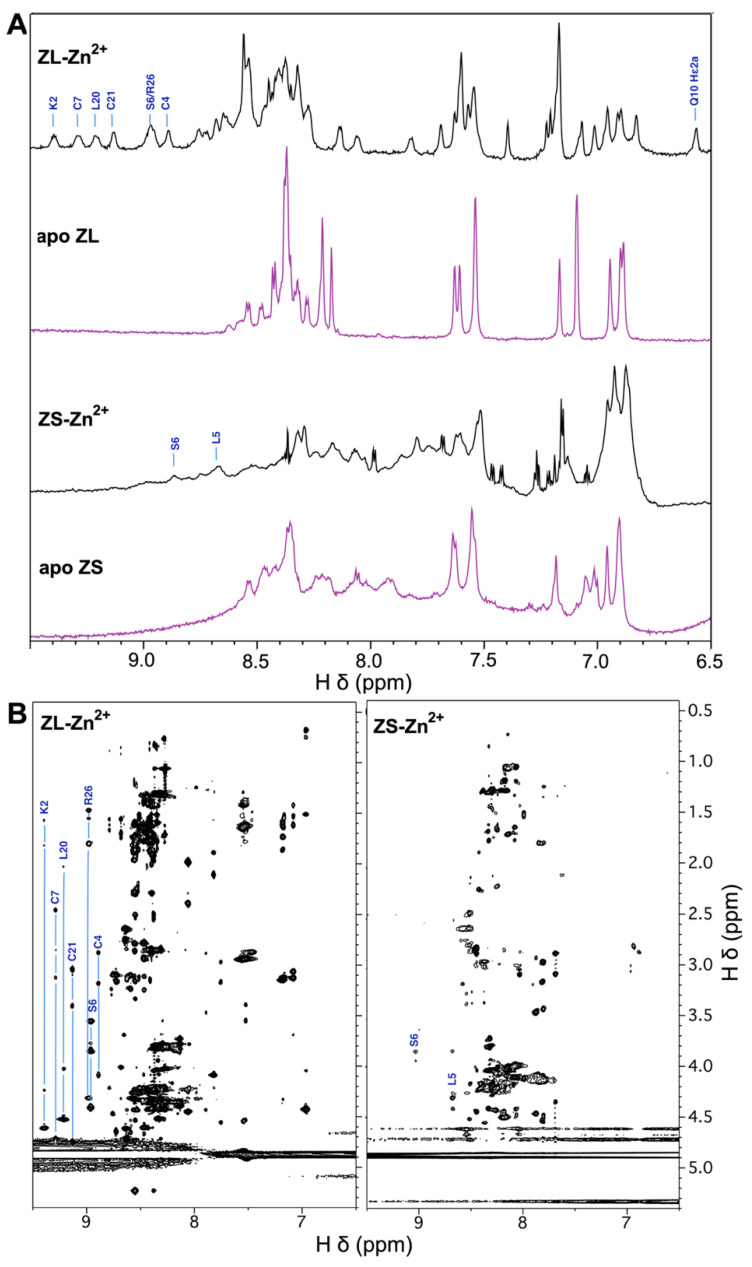
Comparisons of NMR spectra for the ZS and ZL peptide fragments. (**A**) 1D ^1^H-NMR spectra of ZS and ZL in the absence and presence of equimolar ZnCl_2_. (**B**) 2D-TOCSY (70 ms mixing time) of ZS and ZL both with equimolar Zn^2+^. NMR spectra were collected at 800 MHz on 1.6 mM peptide samples at pH 6.0 and a temperature of 10 °C. Selected sequence-specific assignments are given for well-dispersed NMR signals based on the complete assignments for zinc-bound ZL (Appendix A). Note the poorer NMR dispersion for ZS, which has random-coil-like chemical shift ranges (HN ~8.7–7.7 ppm and Hα ~4.6–4.0 ppm).

**Figure 3 biomolecules-15-01091-f003:**
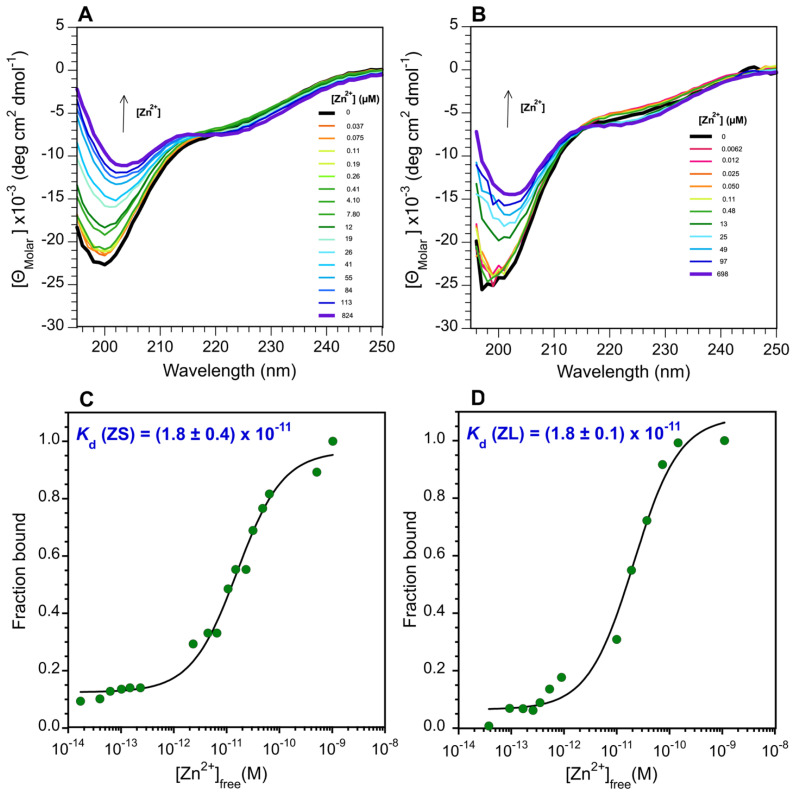
Zn^2+^ binding by CD spectroscopy. ZnCl_2_ titrations with peptide fragments ZS (**A**) and ZL (**B**). CD experiments were performed at a temperature of 20 °C on 80 µM peptide samples in 5 mM HEPES buffer pH 6.8–7.0, containing 0.2 mM of the reducing agent TCEP, 10 mM of the competitive chelator EGTA, and the indicated ZnCl_2_ concentrations. Fits of the binding data as previously described [42,49], were used to obtain *K*_d_ values for Zn^2+^-binding to the ZS (**C**) and ZL (**D**) fragments. Note the logarithmic x-axes in panels C and D. Reported *K*_d_ values are the mean ± SD from duplicate experiments on separate samples of each of the ZS and ZL fragments.

**Figure 4 biomolecules-15-01091-f004:**
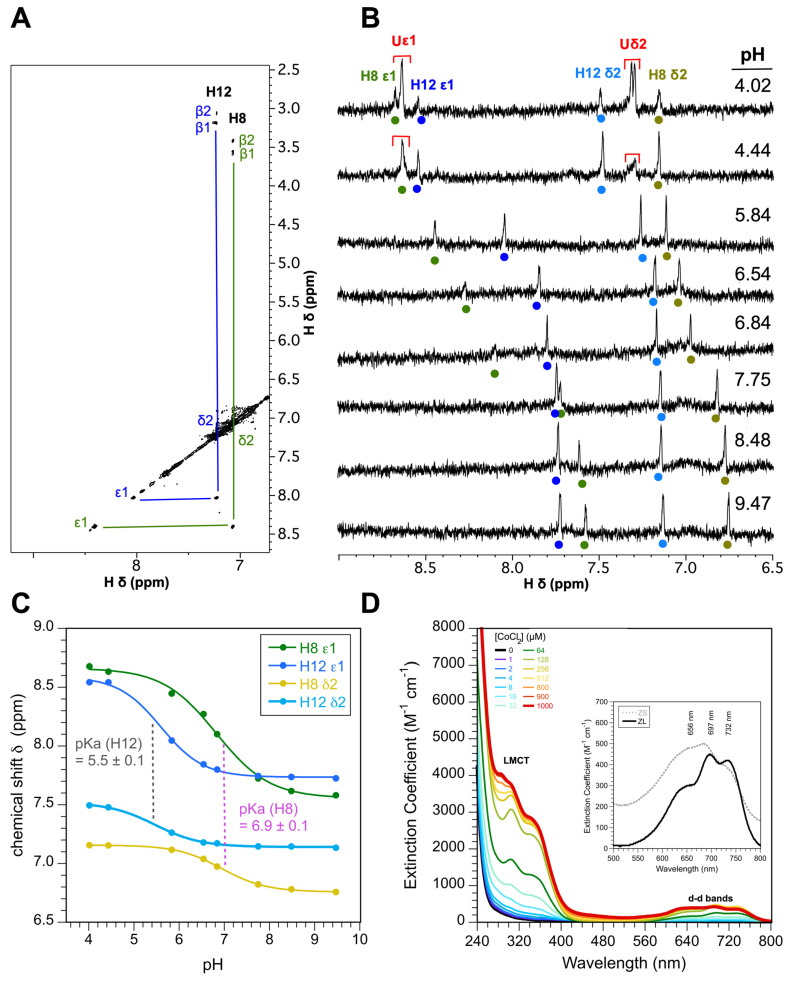
Identification of the metal ligands. (**A**) Assignments of histidine ring protons from ^4^J_Hδ2-Hβ_ couplings in a 70 ms mixing time TOCSY recorded for ZL-Zn^2+^ in D_2_O. (**B**) pH titration of the H8 and H12 aromatic ring resonances. Note that the two histidines are the only aromatic residues in ZL. The fact that both histidines shift with pH indicates that they do not participate in metal coordination. The signals Uε1 and Uδ2 (red) are from the histidines in the acid-unfolded state. (**C**) Titration curves for the two histidines in the folded state and p*K*_a_ values (mean ± SD) calculated from the Hδ2 and Hε1 curves for each histidine. (**D**) Titration of ZL with CoCl_2_. The LMCT (315 nm) and d-d bands (500–800 nm) are consistent with coordination of the metal by four cysteines [60]. The inset compares the d-d band region for ZL (black line) and ZS (dotted gray line), both in the presence of 128 µM CoCl_2_. The similarity of the spectra indicates similar metal coordination sites for ZS and ZL.

**Figure 5 biomolecules-15-01091-f005:**
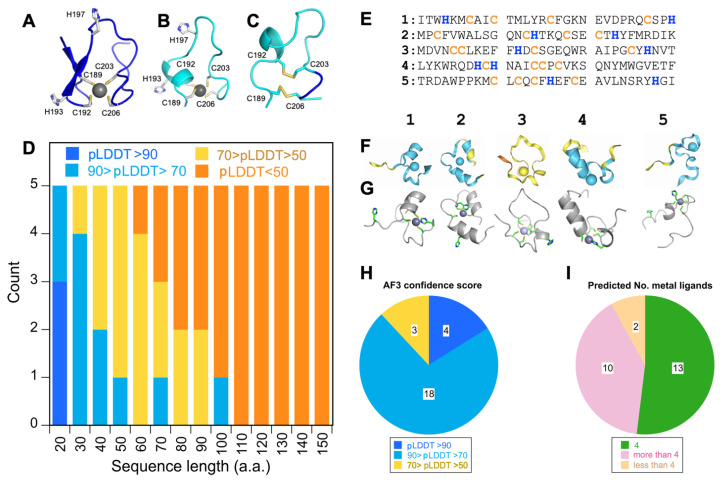
Alpha Fold 3 (AF3) predictions. (**A**) ZL fragment with Zn^2+^. (**B**) ZS fragment with Zn^2+^. Note the difference in metal coordination of C189 and C192 compared to ZL in (**A**). (**C**) The ZS fragment without metal. (**D**) Test of the dependence of AF3 confidence (LDDT score) on sequence length for random sequences. (**E**) Five representative random sequences constrained to have a total of 30 a.a. with four cysteines (yellow) and two histidines (blue), the same distribution of metal-ligating residues as ZC4H2. (**F**) AF3 predictions for the five random sequences in (**E**), colored according to AF3 pLDDT confidence scores. Most of the random sequence predictions have the same confidence levels as the AF3 predictions for the ZNF from ZC4H2 in (**A**–**C**). (**G**) AF3 predictions from panel F showing the cysteine and histidine residues. In all cases four Cys/His residues form a coordination site for Zn^2+^. To obtain better statistics we simulated a larger set of 25 random sequences, each with 27 residues like ZC4H2-ZL and 4 Cys and 2 His. The pLDDT confidence levels for the random sequence predictions (**H**) and the number of cystine and histidines bound to Zn^2+^ (**I**).

**Figure 6 biomolecules-15-01091-f006:**
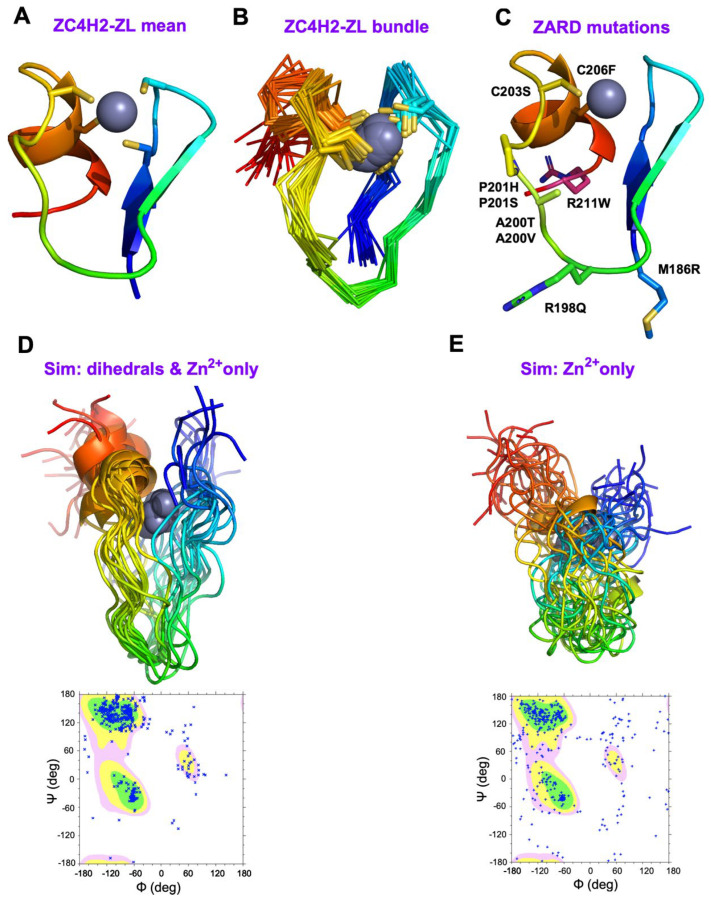
NMR structure of ZC4H2-ZL. The protein mainchain is shown on a color gradient from the N-term (blue) to the C-term (red). Sidechains of the four chelating cysteine residues are shown with sticks, and the zinc atoms as gray spheres. (**A**) NMR structure closest to the ensemble mean. (**B**) The NMR bundle. (**C**) Locations of nine known ZARD mutations in the ZnF, labeled according to the numbering scheme for the full-length ZC4H2 protein. (**D**) Simulated structures calculated with only dihedral and zinc ion restraints, and the associated Ramachandran plot. (**E**) Simulated structures calculated with only restraints to Zn^2+^ and the associated Ramachandran plot. In the Ramachandran plots green, yellow, and pink contours are favored, allowed, and generously allowed regions, respectively. For each structure bundle the 20 lowest-energy conformers were included. Backbone atoms of all residues were superposed, giving RMSD values of 0.69 Å for panel B, 3.9 Å for panel (**D**), and 4.7 Å for panel (**E**).

## Data Availability

Chemical shifts for ZL-Zn^2+^ were deposited in the BioMagResBank under accession number 53214. Restraints and coordinates for NMR structures were deposited in the Protein Data Bank under accession ID 9P3Z.

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
