# Peer review of "Zinc-Induced Folding and Solution Structure of the Eponymous Novel Zinc Finger from the ZC4H2 Protein"

_biomolecules, 2025, doi:10.3390/biom15081091_

Round 1

Reviewer 1 Report

Comments and Suggestions for Authors

This is a very nice and interesting work. In this manuscript the authors explored the structure of a ZF segment from ZC4H2 protein which is related to many neurological disorders. The use two peptides and using variety of NMR and CD experiments, got a lot of information on the structure and zinc coordination. Moreover, based on these experiments the effect of points mutations on the folding and the interaction with DNA or RNA was better understood. The manuscript is well written and in my opinion all experiments were done properly. Therefore, I recommend accepting it to publication.

Author Response

Reviewer 1:

This is a very nice and interesting work. In this manuscript the authors explored the structure of a ZF segment from ZC4H2 protein which is related to many neurological disorders. The use two peptides and using variety of NMR and CD experiments, got a lot of information on the structure and zinc coordination. Moreover, based on these experiments the effect of points mutations on the folding and the interaction with DNA or RNA was better understood. The manuscript is well written and in my opinion all experiments were done properly. Therefore, I recommend accepting it to publication.

Submission Date

21 June 2025

Date of this review

27 Jun 2025 08:41:12

We thank Reviewer 1 for their positive and encouraging review of our manuscript. There were no specific points to address for Reviewer 1.

Reviewer 2 Report

Comments and Suggestions for Authors

This manuscript presents a comprehensive structural characterization of the zinc finger domain of ZC4H2 protein, which is associated with neurodevelopmental and musculoskeletal disorders. While AlphaFold 3 computational predictions provided ambiguous structural models, experimental approaches—including UV-Vis, CD and NMR analysis—confirmed that ZC4H2 contains a zinc-binding motif with picomolar affinity and 4Cys bound to the metal ion. The domain adopts a novel fold reminiscent of RANBP2-type zinc fingers but lacks the conserved residues required for canonical substrate binding. Finally, sequence homology analyses suggest a potential role in RNA binding or alternative molecular functions.

There some points needing to be clarified and expanded:

  1. The authors used model peptides for spectroscopic analysis, the peptide sequences used in this study should be included in the manuscript or in the supplementary material. It is not clear if the used peptide were protected at the N- and C-terminal site and the rationale used for adding extra residues to the wild type sequence.
  2. The observation that amide proton signals broaden and eventually disappear at non-acidic pH is a common behavior of flexible and disordered peptides. Rather than attributing this to thermal instability, it would be more appropriate to consider that such behavior likely reflects the intrinsic flexibility of the peptide, especially given its short length (approximately 20–30 residues). This interpretation is further supported by the spectra obtained for the shorter peptide (ZS), which display similar characteristics. I recommend including a more detailed discussion of this aspect in the manuscript, supported by appropriate references from the literature.
  3. The title of Figure 4, “Identification of the four cysteines as metal ligands”, is misleading, as the data presented in the figure do not directly address or confirm the involvement of cysteine residues in metal coordination. The title should be revised to more accurately reflect the content of the figure, which focuses on the analysis of histidine proton chemical shifts and their potential involvement in zinc binding.
  4. The conclusion that histidines do not participate in zinc coordination is based solely on the pH-dependent chemical shift changes of the aromatic protons of histidine residues. However, this approach alone is not fully convincing. To strengthen this conclusion, the data obtained for the ZL peptide in the presence of Zn(II) (Figures 4B and 4C) should be directly compared with those acquired in the absence of the metal, as the pKa values of histidine residues are expected to shift upon metal binding (see for examples J. Phys. Chem. B 2013, 117, 30, 8954–8965, J. Am. Chem. Soc. 2004, 126, 8, 2602–2612). Furthermore, the authors do not comment on whether the observed histidine signals are shifted or remain unchanged upon zinc addition. This information is critical to definitively rule out histidine involvement in zinc coordination.
  5. The caption for Figure 6 contains a labeling error—panel D is mentioned twice. Please revise the caption to ensure that each panel is uniquely and correctly identified.
  6. The caption for Figure 6 should provide more detailed information regarding panel B. Specifically, the number of structures shown in the NMR ensemble should be indicated, along with the region of the structure used for the fitting (e.g., backbone atoms, specific residues). Additionally, the RMSD value calculated for the ensemble should be reported to give a clearer sense of the structural convergence.
  7. The authors should explain the rationale behind the use of cobalt as a substitute for zinc in their experiments. While this approach is common in studies of zinc-binding proteins due to the spectroscopic advantages of cobalt, the manuscript should explicitly justify this choice and include appropriate references to support the methodology.

Author Response

Reviewer 2:

This manuscript presents a comprehensive structural characterization of the zinc finger domain of ZC4H2 protein, which is associated with neurodevelopmental and musculoskeletal disorders. While AlphaFold 3 computational predictions provided ambiguous structural models, experimental approaches—including UV-Vis, CD and NMR analysis—confirmed that ZC4H2 contains a zinc-binding motif with picomolar affinity and 4Cys bound to the metal ion. The domain adopts a novel fold reminiscent of RANBP2-type zinc fingers but lacks the conserved residues required for canonical substrate binding. Finally, sequence homology analyses suggest a potential role in RNA binding or alternative molecular functions.

We thank the reviewer for their careful reading of the manuscript and overall positive comments.

There some points needing to be clarified and expanded:

  1. The authors used model peptides for spectroscopic analysis, the peptide sequences used in this study should be included in the manuscript or in the supplementary material. It is not clear if the used peptide were protected at the N- and C-terminal site and the rationale used for adding extra residues to the wild type sequence.

We gave the peptide sequences in Fig. 1B. The peptides were also described in Section 2.1 of the Methods and Section 3.1 of the Results. In Section 2.1 we noted that both peptides had acetylated N-termini and amidated C-termini and explained why (to avoid the introduction of charges from free ends thus better mimicking the fragments in the context of the intact protein). Both sequences we looked at were segments of the WT ZC4H2 protein. We noted and corrected a mistake in the domain boundary given for the ZL fragment in Section 3.1 that might have caused confusion. In section 3.1, we tried to better clarify the choice of peptide fragments (also in light of comment #2 of Reviewer 3).

  1. The observation that amide proton signals broaden and eventually disappear at non-acidic pH is a common behavior of flexible and disordered peptides. Rather than attributing this to thermal instability, it would be more appropriate to consider that such behavior likely reflects the intrinsic flexibility of the peptide, especially given its short length (approximately 20–30 residues). This interpretation is further supported by the spectra obtained for the shorter peptide (ZS), which display similar characteristics. I recommend including a more detailed discussion of this aspect in the manuscript, supported by appropriate references from the literature.

We apologize for the confusion but the NMR line broadening effects we observe are not due to amide proton exchange. We showed the amide region of the spectrum since it had better dispersion due to the lack of aromatic residues (other than His) in the peptides. The data in Fig. 2A were at pH 6.0 and 10 C, conditions close to the minimum for hydrogen exchange, and indeed we see amide protons even for the unfolded peptides without zinc under these conditions. To clarify this point we included the aliphatic regions of the 1D spectra corresponding to Fig. 2A in a new supplementary Figure S3.

  1. The title of Figure 4, “Identification of the four cysteines as metal ligands”, is misleading, as the data presented in the figure do not directly address or confirm the involvement of cysteine residues in metal coordination. The title should be revised to more accurately reflect the content of the figure, which focuses on the analysis of histidine proton chemical shifts and their potential involvement in zinc binding.

We changed the figure title to “Identification of the metal ligands”

  1. The conclusion that histidines do not participate in zinc coordination is based solely on the pH-dependent chemical shift changes of the aromatic protons of histidine residues. However, this approach alone is not fully convincing. To strengthen this conclusion, the data obtained for the ZL peptide in the presence of Zn(II) (Figures 4B and 4C) should be directly compared with those acquired in the absence of the metal, as the pKa values of histidine residues are expected to shift upon metal binding (see for examples J. Phys. Chem. B 2013, 117, 30, 8954–8965, J. Am. Chem. Soc. 2004, 126, 8, 2602–2612). Furthermore, the authors do not comment on whether the observed histidine signals are shifted or remain unchanged upon zinc addition. This information is critical to definitively rule out histidine involvement in zinc coordination.

In addition to the histidine titration data, the UV-VIS LMCT e315 extinction coefficient of about 4000 M-1 cm-1 in Fig. 4D supports 4 cysteines binding to Co2+, implying that the histidines are not involved in the tetrahedral metal coordination. The experimental NMR structure as well as the AlphaFold3 prediction both have the four cystines coordinating zinc, with no involvement from the histidines in metal binding. We (PMID: 39180464) and others (PMID: 9049312) have observed in several zinc fingers that histidines coordinating zinc are insensitive to pH (over a range from about pH 4.5 to 10), since metal binding resists protonation. We therefore concluded that since the two histidines in ZC4H2 titrate freely, they do not bind zinc. Of course, titration of a nearby group could lead to pH-dependent changes in the NMR spectrum, but there are no other groups in the ZL peptide that titrate with a pKa ~6 to 7 typically of histidines. The reference cited by Reviewer 2 (J. Phys. Chem. B 2013, 117, 30, 8954–8965) describes a ssNMR study of Zn complexed by the amino acid histidine (a considerably different system than Zn coordinated by a ZnF peptide). In the ssNMR study, pH-induced changes in Zn coordination involve deprotonation of the His Ne2 atom and loss of metal coordination of the alpha amino group (which is not free in our polypeptide). These changes occur at pH 11 and pH 14, well outside the pH range we studied. The second paper (J. Am. Chem. Soc. 2004, 126, 8, 2602–2612) is a theoretical study on histidine protonation in catalytic Zn-His-Asp/Glu triads that is not straightforward to relate to the present study. For completion, we did measure pKa values for the two histidines in the absence of Zn and include these in a new Figure S5. Although we do not have NMR assignments for the two histidines in unfolded ZL the pKa values of 6.0 are the same for both histidines. The chemical shifts and pKa values change between the unfolded and folded state (the chemical shift differences can also be seen in the pH 4.0 spectrum of Fig. 4B in the main text). This is not because of metal-bonding by the histidines but because of the folding of the peptide accompanying zinc binding. Changes in chemical shifts and pKa values of histidines accompanying folding have been demonstrated for numerous proteins that do not bind metals (PMID: 3288282, 17920624, and 27006399 are some examples).

  1. The caption for Figure 6 contains a labeling error—panel D is mentioned twice. Please revise the caption to ensure that each panel is uniquely and correctly identified.

Fixed. Thank you for catching this.

  1. The caption for Figure 6 should provide more detailed information regarding panel B. Specifically, the number of structures shown in the NMR ensemble should be indicated, along with the region of the structure used for the fitting (e.g., backbone atoms, specific residues). Additionally, the RMSD value calculated for the ensemble should be reported to give a clearer sense of the structural convergence.

We included all the information in the figure 6 legend

  1. The authors should explain the rationale behind the use of cobalt as a substitute for zinc in their experiments. While this approach is common in studies of zinc-binding proteins due to the spectroscopic advantages of cobalt, the manuscript should explicitly justify this choice and include appropriate references to support the methodology.

We improved the description of the cobalt experiments and added appropriate references in the last paragraph of Section 3.2.

Submission Date

21 June 2025

Date of this review

07 Jul 2025 23:22:22

Reviewer 3 Report

Comments and Suggestions for Authors

This manuscript describes the biophysical and structural characterization of a zinc finger domain which was previously not confirmed yet. Authors clearly demonstrated that this domain formed a zinc finger upon binding to the zinc ion through their comprehensive approach including multiple spectroscopic techniques such as NMR, CD, UV-vis. The most impressing was the comparison with the AlphaFold. This stresses that the prediction should experimentally be verified especially for short peptides. I have only a couple of minor comments.

  1. The purity of ZS was 90% while ZL was 97%. Wouldn't it be better if authors explicitly mentioned that there was no effect of peptide purity on folding or structure?
  2. UniProt predicted residues 189-206 for zinc finger. How did authors define the residues for ZS or ZL?

Author Response

Reviewer 3:

This manuscript describes the biophysical and structural characterization of a zinc finger domain which was previously not confirmed yet. Authors clearly demonstrated that this domain formed a zinc finger upon binding to the zinc ion through their comprehensive approach including multiple spectroscopic techniques such as NMR, CD, UV-vis. The most impressing was the comparison with the AlphaFold. This stresses that the prediction should experimentally be verified especially for short peptides. I have only a couple of minor comments.

We thank Reviewer 3 for their positive review of our manuscript. Specific points are addressed below.

  1. The purity of ZS was 90% while ZL was 97%. Wouldn't it be better if authors explicitly mentioned that there was no effect of peptide purity on folding or structure?

We ordered peptides at >90% purity, but ZL was purified to 97% so we listed the HPLC purities reported by the manufacturers. We agree with the reviewer that the difference in purity does not affect folding or structure, since the CD spectra of the peptides with zinc and the binding constants for zinc are similar. The changes in NMR spectra between the two peptides are indicative of differences in dynamics.

  1. UniProt predicted residues 189-206 for zinc finger. How did authors define the residues for ZS or ZL?

We explained this in section 3.1 of the original manuscript but tried to make this clearer in the revised version. We first looked at the ZS fragment (188-207) that included one extra residue on either side of the boundaries given by UniProt (189-206). This fragment gave poor NMR spectra suggesting it was too short (but still bound zinc). In particular the ZS fragment has its N-terminus in the middle of the first b-strand predicted by AF3, or the homologous RANBP2 structures in Table 1C. We therefore made a longer ZL fragment (186-212) that extends both the N- and C-terminus. In the ZL fragment the N-terminus is past the beginning of the predicted strand b1, and an extra stretch of 5 polar residues at the C-terminus (compared to ZS) is likely to improve solubility. The longer ZL fragment (186-212) gives much better NMR spectra than the shorter ZS fragment (188-207), even though both fragments are longer than the domain boundaries specified by UniProt (189-206).

Round 2

Reviewer 2 Report

Comments and Suggestions for Authors

The authors adressed all the comments